# The Regulatory Networks of the Circadian Clock Involved in Plant Adaptation and Crop Yield

**DOI:** 10.3390/plants12091897

**Published:** 2023-05-06

**Authors:** Hang Xu, Xiling Wang, Jian Wei, Yi Zuo, Lei Wang

**Affiliations:** 1Key Laboratory of Plant Molecular Physiology, Institute of Botany, Chinese Academy of Sciences, Beijing 100093, China; 2University of Chinese Academy of Sciences, Beijing 100049, China; 3College of Life Sciences, Changchun Normal University, Changchun 130032, China

**Keywords:** circadian clock, crop, plant adaptation, yield

## Abstract

Global climatic change increasingly threatens plant adaptation and crop yields. By synchronizing internal biological processes, including photosynthesis, metabolism, and responses to biotic and abiotic stress, with external environmental cures, such as light and temperature, the circadian clock benefits plant adaptation and crop yield. In this review, we focus on the multiple levels of interaction between the plant circadian clock and environmental factors, and we summarize recent progresses on how the circadian clock affects yield. In addition, we propose potential strategies for better utilizing the current knowledge of circadian biology in crop production in the future.

## 1. Introduction

As it revolves around the sun, the rotation of the Earth results in a combination of diel and seasonal changes to environmental factors such as light intensity, day length, and temperature. In order to adapt to these continual yet predictable environmental changes, the endogenous time-keeping mechanism in higher plants, known as the circadian clock, has evolved [1,2]. Transcriptomic studies show that the circadian clock system regulates nearly 80% of all genes in rice (*Oryza sativa*), poplar (*Populus trichocarpa*), and Arabidopsis (*Arabidopsis thaliana*) [3]. Schematically, plant circadian clocks consist of an integrated system of input, core oscillator, and output pathways. The inputs include external signals, such as light and temperature, which influence the pace of the circadian clock, entraining it by impinging on different molecular processes at the core oscillator. The core oscillator consists of a few interlocked transcription-translation feedback loops. In *Arabidopsis,* levels of *CIRCADIAN CLOCK ASSOCIATED 1* (*CCA1*) and *LONG ELONGATED HYPOCOTYL* (*LHY*), two MYB-like transcription factors, peak at dawn. Their encoded proteins repress the transcription of *PSEUDO-RESPONSE REGULATOR* (*PRR*) gene family members [4]. In turn, the PRR protein family, including PRR9, PRR7, PRR5, and TIMING OF CAB EXPRESSION 1 (TOC1), which are sequentially expressed from morning to dusk, inhibit the transcription of *CCA1* and *LHY*, forming the core feedback loop of the circadian core oscillator [5,6]. The expression of *PRR genes* and *LUX ARRHYTHMO* (*LUX*) can be repressed by the evening complex (EC), which consists of EARLY FLOWERING 3 (ELF3), EARLY FLOWERING 4 (ELF4), and LUX [7].

Recently, it was found that many staple crops share evolutionarily conserved core oscillator genes with the model plant Arabidopsis [8]. In particular, the homologs of Arabidopsis *CCA1* and *LHY*, along with *GIGANTEA* (*GI*), function in both eudicot and monocot plants [8]. Consistently, the circadian clock regulates crop yields on a variety of levels, including plant physiology, biochemistry, development, and metabolic processes. Here, we summarize recent findings about how the circadian clock responds to varying environmental light and temperature cues, as well as abiotic and biotic stresses, to ensure optimal plant growth and reproduction. Based on evidence that the circadian clock affects many agriculturally important traits and enhances their environmental adaptability [2], we propose that a better understanding of the clock will generate strategies to significantly increase crop yields and help us to meet the needs of our growing global population.

## 2. The Role of the Circadian Clock in Controlling Photoperiodic Flowering Time

The widespread distribution of plants across multiple latitudes has largely depended on the evolution of endogenous circadian clock systems that sense and coincide with external photoperiods. When taken together, the circadian clock system and light signals coordinately sense photoperiodic information to determine the proper flowering time [9]. Crop growth and yields are also, to a large extent, determined by sunlight, depending on the close interaction between the circadian clock and light signals. Light signals are perceived by a variety of plant photoreceptors, which sense light quality and quantity, and are integrated by the circadian clock system to control downstream output pathways, such as those for gene expression and physiological responses [10]. Specifically, the downstream output pathways, including flowering time, photosynthetic production, and nutrient absorption, collectively influence and determine crop yields (Figure 1). In order to ensure sufficient agricultural production in the future, it will be very important to expand the cultivation area of elite crop cultivars. Thus, one of the most important current issue questions is: how can crops best be adapted to new photoperiodic conditions when grown across a wide range of latitudes?

The sensors that sense light signals and establish links with the circadian clock are known as photoreceptors. Arabidopsis processes five types of photoreceptors: phytochromes, cryptochromes, phototropins, Zeitlupe (ZTL), and UV RESISTANCE LOCUS 8 (UVR8) [11] (Figure 2A). Phytochromes are red and far-red light sensors, including phytochromes A to E. Interestingly, cryptochromes (CRY1 and CRY2), the two main UV-A and blue light photoreceptors, are evolutionarily conserved in plant, fungi, bacteria, and animals [12]. The phototropins Phot1 and Phot2are also blue light photoreceptors [11]. Unlike other photoreceptors, which are located in the nucleus, phototropins are mainly found in the plasma membrane. In contrast to other photoreceptors, phototropins do not influence the expression of circadian genes in the nucleus but instead, maintain the robust circadian rhythms of photosystem II operating efficiency in response to low levels of blue light [13]. Members of the Zeitlupe photoreceptor family, including ZTL1, LOV KELCH PROTEIN 2 (LKP2), and FLAVIN-BINDING KELCH REPEAT F-BOX PROTEIN1 (FKF1), mainly absorb UV-A/blue light [14]. Finally, UVR8 is a UV-B photoreceptor that also plays an important role in the entrainment of the circadian clock [15].

The regulatory networks controlling Arabidopsis flowering time provide a general mechanistic model for photoperiodic flowering. The circadian system controlling the Arabidopsis flowering time pathway mainly involves CONSTANS (CO) and FLOWERINGLOCUS T(FT) [16]. FT, as a florigen, initiates the flowering process under long-day conditions in Arabidopsis [17]. FT protein is a phospholipid-binding protein belonging to the phosphatidyl ethanolamine binding protein (PEBP) family, which is highly conserved in numerous plant species [18,19]. FT is produced in leaves and migrates to the apical meristem, where it functions as a remote signal [20]. FT is a floral pathway integron, and its expression is finely regulated by several transcription factors, including CO, its primary activator; FT was the first member of the plant-specific B-box transcription factor family [21,22]. The level of CO determines the magnitude of induction of *FT* expression. A recent structural analysis found that CO regulates *FT* expression via its C-terminal CONSTANS, CONSTANS-LIKE, and TOC1 (CCT) domain [23]. The connection between the circadian clock and photoperiodic flowering is established mainly through the regulatory action of CO, including both transcriptional and post-translational regulation.

In addition, Arabidopsis CYCLING DOF FACTOR 1 (CDF1) to CDF5 regulate CO by functioning as transcriptional repressors [24] (Figure 2A). Members of the CDF protein family inhibit the transcription of CO by binding to its promoter in the morning [25]. The expression of CDF is regulated by many circadian clock components. For instance, CCA1 and LHY promote the transcription of CDF in the morning, whereas PRR9, PRR7, and PRR5 suppress its transcription in the afternoon [26] (Figure 2A). Additionally, PRR proteins physically interact with CO and then stabilize it in a time-specific fashion, thus mediating CO accumulation under long-day conditions [27]. The GI and KELCH REPEAT, F-BOX 1 (FKF1) proteins form a complex that is blue light-dependent and regulates CO transcription. When FKF1 interacts with the GI-CDF1 complex in the afternoon, FKF1 can degrade CDF1, releasing the inhibition of CO [28] (Figure 2A). The phosphorylation of key circadian clock proteins is also involved in the control of flowering [29]. Studies have shown that CASEIN KINASES 2(CK2) phosphorylates CCA1 to ensure its DNA binding capacity to the *LIGHT-HARVESTING CHLOROPHYLL A/B1*3* (*LHCB1*3*) promoter [30].

In contrast to the long-day plant Arabidopsis, short-day plants like soybean (*Glycine max*) and rice possess unique flowering regulatory networks. Soybean is a typical short-day crop, with short-day conditions promoting its flowering and long-day conditions prolonging its vegetative growth. The deletion of soybean *GmLCLa1*, a CCA1/LHY homologous gene, along with *LCLa2*, *LCLb1*, and *LCLb2*, results in an extreme short-period circadian rhythm and late-flowering phenotype [31]. A recent study identified *Juvenile*(*J*)as an ortholog of Arabidopsis *ELF3*, the major locus conferring the long-juvenile trait, which prolongs the vegetative phase and improves yield under short-day conditions. The vegetative growth period of a soybean *j* mutant in southern China was significantly prolonged, and its flowering time was delayed, thus increasing the final yield by providing more photosynthetic products before flowering [32]. Two soybean homeologs of LUX interact with J to form soybean EC. Accordingly, soybean plants lacking *LUX1* and *LUX2* showed an extremely late-flowering phenotype under both long- and short-day conditions, suggesting that the circadian clock EC may regulate soybean sensitivity to photoperiod [33] (Figure 2B). In contrast, the mutation of soybean *Timing of flowering 12* (*Tof12*) and *Tof11*, two genes homologous to *PRR3*, enabled cultivation at higher latitudes by ensuring proper flowering time and high crop yield [34]. In addition, the soybean phytochromes PHYA2 and PHYA3 directly interact with LUX proteins, reducing their stability and promoting *E1* expression and late flowering. Intriguingly, PHYA3 and PHYA2 may also interact with E1 to stabilize it [35] (Figure 2B).

Although rice, like soybean, is a short-day crop, it has a different regulatory network controlling photoperiodic flowering time. The Arabidopsis CO-FT pathway of the flowering regulation pathway is conserved in rice, where it is known as the Heading date 1 (Hd1)-Hd3a/ RICE FLOWERING LOCUS T 1 (RFT1) module. In particular, there is a unique route for the suppression of flowering via long days in rice, involving the proteins Grain Number, Plant Height and Heading Date 7 (Ghd7), Early Heading Date 1 (Ehd1), Hd3a, and RFT1 [36]. The rice clock component OsCCA1 also plays a vital role in the control of heading date control. The loss of *OsCCA1* results in a late flowering phenotype, which indicates that OsCCA1 is a floral activator. OsCCA1 is reported to bind to the *OsPRR37* promoters to repress the expression of *OsPRR37*, leading to the up-regulation of florigens [37] (Figure 2C). In the Nipponbare cultivar of the japonica rice subspecies, the clock gene *OsPRR37* enhances photoperiod sensitivity regarding the regulation of heading date (flowering time in rice) [38].

Casein kinases 1 (CK1) and CK2α interact with OsPRR37 and subsequently phosphorylate the different amino acid residues of OsPRR37, thereby affecting the activity of the protein [39]. In rice, there are five PRR family genes: *OsPRR37*, *OsPRR73*, *OsPRR59*, *OsPRR95* and *OsPRR1* [40] (Table 1). OsPRR37 encodes a protein that regulates the adaptability of rice to different latitudes, with the variation in *OsPRR37* potentially expanding the area of adaptability and making a contribution to rice cultivation [41]. OsPRR37 directly suppresses the transcription of the day-phased clock genes *OsCCA1*, *OsPRR73*, and *OsPRR95* and indirectly enhances the expression of the evening-phased clock genes *OsPRR1* and *OsPRR59* [42]. OsPRR37 and OsPRR73 show partial functional redundancy in the regulation of rice salt stress responses at the panicle stage and are also involved in regulating heading date [42,43]. *OsPRR73*-overexpressing plants show a late flowering phenotype under both long-day and short-day conditions, perhaps because the binding of OsPRR73 to the promoters of *Ehd1* and *OsCCA1* inhibits their expression at dawn [43]. *OsPRR59* and *OsPRR95* are directly involved in the regulation of photosynthesis, and their expression is also regulated by photoperiod [44].

Rice contains two *ELF3* orthologs, *OsELF3-1* and *OsELF3-2*; of these, OsELF3-1 plays a more significant role in controlling the heading date of rice. Under short-day conditions, OsELF3-1 advances the heading date by activating *Ehd1* but acts as an inhibitory factor under long-day conditions (Figure 2C). In addition, OsELF3-1 inhibits the expression of other clock genes, such as *OsPRR1*, *OsPRR37*, *OsPRR73*, and *OsPRR95* [45]. OsEC1, a protein complex consisting of OsELF4a, OsELF3-1, and OsLUX, directly binds to the promoter of *OsGI*, repressing its expression [46], whereas *OsGI* acts upstream of *Hd1* to determine the heading date. The knocking down of *OsGI* expression results in an early flowering phenotype in long-day conditions and a late flowering phenotype in short-day conditions [47]. In addition, OsGI can interact with Ghd7 to induce the degradation of Ghd7, whereas OsPHYA and OsPHYB suppress the interaction of OsGI and Ghd7, thus stabilizing Ghd7 [48] (Figure 2C).

In addition, CCA1/LHY, PRRs, GI, ELF3, and ELF4 were shown to function in photoperiod-dependent flowering regulation in other crops, including soybean, pea (*Pisum sativum* L.), Broccoli (*Brassica oleracea* L. var. *italica*), wheat (*Triticum aestivum* L.), and maize (*Zea mays*)(Table 1). However, how light signals and the circadian clock coordinately regulate flowering time in crops remains to be further explored, both biochemically and genetically.

## 3. Interplay between the Circadian Clock and Temperature Cues

The plasticity of plant circadian systems means they can be entrained and reset by environmental signals, such as temperature changes, which ensures better adaptation to an ever-changing environment [49,50,51] (Figure 1). With the intensification of global warming, an understanding of how to utilize the plasticity of crop circadian clocks is crucial for optimal crop yield and food security. Furthermore, the expansion of grain production areas to higher-latitude regions requires adaptation to a colder environment. Therefore, to improve crop yield, understanding the interaction and communication mechanisms between circadian clocks and temperature signals is becoming increasingly important and urgent. Moreover, the circadian clock is subject to temperature compensation, whereby the circadian period maintains a relatively steady state in spite of temperature changes [50,51]; however, how the circadian clock systems of crops compensate for and are reset based on environmental temperature fluctuations remains poorly understood.

In Arabidopsis, there are many studies that suggest a comprehensive interaction between the circadian clock system and temperature signals. Several circadian clock genes, including *CCA1*, *LHY*, *GI*, *PRR7*, *PRR9*, and the genes encoding EC components, have been shown to function in temperature compensation [52,53,54]. For example, ELF3, a member of the EC, functions as a thermosensor. At higher temperatures, ELF3 fused with green fluorescent protein forms speckles within a few minutes in response to higher temperatures. Hence, it can quickly switch from an active to an inactive state via phase transitions induced by the warmer temperature [55]. Moreover, recent studies revealed that PIF4 and ELF3 could retain short-term memory of the daytime temperature condition [56], indicating that the circadian clock has additional complex relationships with temperature signals. Besides, a high temperature could also enhance both CCA1 binding capacity and CK2 phosphorylation to maintain a stable circadian period among temperature fluctuations, suggesting a post-translational modification mechanism underlying temperature compensation [57].

Under mild-to-warm-temperature conditions, higher plants can undergo a series of morphological changes, such as hypocotyl and petiole elongation and leaf hyponasty, in a process known as thermo-morphogenesis [58]. Studies on thermo-morphogenesis have focused on the molecular mechanisms of plant adaptation to rising ambient temperature that could contribute to increased crop yields due to global warming. PIF4 is a key central regulator in plant thermo-morphogenesis that functions in adaptation to high temperatures, and its transcription efficiency is increased at high ambient temperatures [59]. EC is recruited to the promoter of *PIF4* to repress its expression, but this inhibition decreases when the temperature increases [60] because the ability of its EC to bind to genome-wide targets is temperature dependent [61,62,63,64]. Meanwhile, the plant circadian clock may also function in plant responses to cold stress. Previous studies indicated that Arabidopsis CCA1 and LHY are involved in the cold-inducible expression of *DEHYDRATION-RESPONSIVE ELEMENT BINDING PROTEIN1* (*DREB1*) [65]. Cold stress induces the degradation of CCA1 and LHY, leading to the weaker inhibition of CCA1 and LHY on *DREB1s* to respond to cold stress [66]. In addition, PRR7 binds to the promoter of *DREB1* to respond to low-temperature stress [67]. ELF3 acts as a substrate adaptor between CONSTITUTIVE PHOTOMORPHOGENIC 1 (COP1) and GI to promote GI destabilization in the dark, accelerating the turnover of GI and ELF3 [68]. COP1 acts as an E3 ubiquitin-ligase. The COP1-dependent turnover of GI is enhanced under low temperatures due to increased COP1 stability [69].

Temperature fluctuation triggers the occurrence of alternative splicing (AS), which is a mechanism that generates different mRNA transcripts from a single gene [70]. A recent study on sugarcane found that temperature modulated the daily dynamics of AS forms in the circadian clock genes [71].Five sugarcane clock genes, *ScLHY*, *ScPRR37*, *ScPRR73*, *ScPRR95*, and *ScTOC1*, exist in at least one alternatively spliced isoform. AS isoforms varied according to the season, indicating that AS in clock genes is correlatedwith temperature fluctuation [71].

However, there is so far little evidence about how the circadian clock co-ordinates with temperature signals to ensure high crop yields. We proposed that studies in Arabidopsis are likely to provide a reference point for exploring how crop temperature responses can co-ordinate with circadian clock systems. In the next few decades, ensuring high crop yields under heat stress and expanding grain production areas to higher latitudes will become increasingly important for future food security. Therefore, it is worth exploring how the circadian clock system functions in integrating temperature signals.

## 4. The Circadian Clock Is Involved in Tolerance to Multiple Abiotic Stresses

The circadian clock plays a fundamental role in coordinating the trade-off between plant growth and abiotic stress responses, thereby modulating crop yields under natural conditions. Here, we will summarize the progress made to date in understanding how the plant circadian clock responds to various environmental stresses and how we could take advantage of circadian clock plasticity to improve crop yields.

Circadian clock components function as important integrators of plant salt stress responses. In Arabidopsis, *prr5prr7prr9* triple mutant plants display more tolerance to salt stress than wild-type plants [72]. On the other hand, the relative expression of *PRR7* and *TOC1* is inhibited in a saline environment, whereas the expression of *PRR9* is induced [73]. The Salt-Overly Sensitive (SOS) signaling pathway is intricately involved in circadian clock responses to salt stress in Arabidopsis [74,75] (Figure 3A). After the degradation of GI mediated by the clock component ELF3, the kinase SOS2 was released from inhibition. It then interacts with SOS3 to form an active complex that activates the expression of *SOS1*, which encodes a membrane-localized Na^+^/H^+^-antiporter [76]. Therefore, ELF3 can enhance salt stress tolerance by degrading GI. In addition, SOS1 can directly interact with GI and stabilize it in a salt-dependent manner [77] (Figure 3A). In addition, ELF3 represses *PIF4* transcription, leading to the upregulation of the stress-tolerance genes *DREB2A* and *DELLA* and conferring salt tolerance [78] (Figure 3A). Similarly, soybean *J*, the ortholog of Arabidopsis *ELF3*, can also enhance salt tolerance by positively regulating salt-stress response genes [79].

In rice, the components of the circadian clock have also been reported to play significant roles in salt stress responses. OsPRR73, a core component of the circadian clock of rice, confers salt tolerance by interacting with HISTONE DEACETYLASE10 (OsHDAC10) to repress the transcription of *HIGH-AFFINITY K^+^ TRANSPORTERS2;1* (*OsHKT2;1*), thus regulating cellular Na^+^ homeostasis [80] (Figure 3B). Another clock component, OsCCA1, functions as a key integrator of salt, osmotic, and drought stresses by regulating the expression of genes in the ABA signaling pathway. Specifically, OsCCA1 binds to the promoters of *OsPP2C* (*PROTEIN PHOSPHATASE 2C*) members and *OsbZIP46* (*BASIC REGION AND LEUCINE ZIPPER 46*) to activate their transcription, thus activating a response to salt stress [81] (Figure 3B). In addition, OsEC1, which consists of OsELF4a, OsELF3-1, and OsLUX, also directly binds to the promoter of *OsGI* to repress its expression. This conveys the stress signaling response by affecting the expression of the downstream ion transporters *OsHKT2;1*, *OsHKT 2;3*, *OsHKT 2;4*, enabling a response to salt stress and regulating the heading date [46] (Figure 3B). As expected, plant circadian clock components regulate salt stress responses at different levels, from transcriptional modification to post-transcriptional modification. Therefore, it will be critical to investigate how plant circadian systems respond to salt stress during different growth stages, with special emphasis on exploring the precise molecular mechanisms through which the crop circadian clock regulates salt stress responses.

The resistance to drought stress relies in part on endogenous plant hormones such as abscisic acid (ABA) [82]. ABA is involved in drought resistance primarily by inducing stomatal closure [83]. Studies indicate that leaf ABA content oscillates on a daily basis, rising during the day and decreasing at night [84]. This dynamic ABA level hints at the close relationship between the circadian clock and ABA. The relationship between drought response signaling and the circadian clock has been investigated in a few plant species [82,85]. Under drought stress, Arabidopsis TOC1 binds to the promoters of ABA-related genes, such as *ABA-RECEPTOR* (*ABAR*), also known as *CHELATASE SUBUNIT H* (*CHLH*), and regulates their circadian expression. Moreover, both TOC1’s expression and the timing of its binding to the *ABAR* promoter are affected by ABA conversely. Hence, TOC1 and ABAR provide a finely tuned switch for moderate responses to drought stress [86]. Moreover, the clock component GI also functions as a key gatekeeper of ABA-regulated responses to drought stress due to its specific sensitivity to ABA [87]. Additionally, under drought conditions, LHY enhances the expression of ABA-response genes [88]. In *Arabidopsis* and poplar, the clock component ZEITLUPE (ZTL) also affects ABA-induced responses, causing stomatal closure that reduces water loss under drought stress [89]. In soybean, several *GmLCLs* (*GmLCLa1*, *GmLCLa2*, *GmLCLb1*, and *GmLCLb2*), the orthologs of *AtCCA1* and *AtLHY*, negatively regulate the drought response [90] (Table 1). The soybean *lcl* quadruple mutant promotes stomatal closure and reduces the rate of water loss from leaves under drought stress, thus increasing drought tolerance [91]. In rice, the core clock component OsCCA1 orchestrates ABA signaling to confer drought response signaling [81]. Evidently, there is a complex regulatory network connecting the circadian clock with drought stress responses, and this interaction is largely mediated by ABA levels and signaling.

Because many circadian clock components regulate both growth and stress responses in plants, it is conceivable that they determine the trade-off between growth and abiotic stress defense and thereby enable the plant to adapt to a naturally harsh living environment. Intriguingly, some circadian clock components negatively regulate stress responses, but others function as positive regulators under abiotic stress. The existence of these two seemingly contradictory regulatory mechanisms highlights the significance of the circadian clock. It will be intriguing to explore how to take advantage of our understanding of circadian systems in depth to maximize crop yields by balancing plant growth and defense.

## 5. The Circadian Clock System and Biotic Interactions

The primary function of the circadian clock is to regulate and entrain the circadian rhythms of various physiological and biochemical processes, such as the expression of resistance genes, hormone synthesis and signal transduction, the homeostasis of reactive oxygen species (ROS), and the opening and closing of stomata. This enhances the resistance to biotic stress between day and night (gating) [92,93,94]. The circadian clock, together with the gating response, helps plants to actively prevent pathogen invasion and allow the best allocation of limited resources [95]. It regulates plant immunity by modulating multiple signaling pathways, including pathogen-associated molecular pattern (PAMP)-triggered immunity (PTI) and effector-triggered immunity (ETI), the salicylic acid (SA) and jasmonic acid (JA) responses, and the ROS-dependent stimulatory signaling pathway [96,97,98]. PTI is activated through pattern-recognition receptors at the plasma membrane that detects pathogen molecules, such as bacterial pathogens’ flagellin or fungal chitin [99,100].

ETI is another type of natural immune response triggered in plants, involving the detection of pathogen-encoded virulence factors, also known as effectors [101]. Previous studies have shown that different circadian clock components are involved in defending against attacks by pathogens and pests. For example, to provide immunity against *Magnaporthe oryzae*, the rice blast fungus, *OsELF3-2*, interacts with E3 ligase AVRPIZ-T INTERACTING PROTEIN 6 (APIP6), and OsELF3-2 is then degraded by the ubiquitin proteasome system to negatively regulate PTI immunity [102]. In addition, the circadian clock core component CCA1/LHY impacts ETI by regulating the expression of resistance (*R*) genes to control the timing of the immune response [98]. Meanwhile, CCA1 regulates resistance to the downy mildew pathogen *Hyaloperonospora arabidopsidis* mediated by the R gene *RPP4* by directly binding to the *RPP4* promoter [96]. Thus, the circadian clock system endows plants with strong immune resistance to biotic stress.

SA and JA are two hormones that are vital in plant defense [103]. Interestingly, the biosynthesis pattern of both SA and JA oscillates during daily periods, indicating that the levels of both proteins are under the control of the circadian clock system [104,105]. Moreover, their signaling pathways are also closely regulated by the circadian clock. Arabidopsis *TIME FOR COFFEE* (*TIC*), a key determinant of circadian-gated processes, acts as a negative regulator of JA signaling via the post-translational repression of the accumulation of the transcription factor MYELOCYTOMATOSIS PROTEIN 2-(MYC2) upon bacterial pathogen and fungal infection [106]. In addition, the clock component LUX can bind to the promoters of *ENHANCED DISEASE SUSCEPTIBILITY 1* (*EDS1*) and *JASMONATE ZIM-DOMAIN 5* (*JAZ5*), which play important roles in the JA and SA signaling pathways. JA and its analogs also activate plant defense against herbivorous insects. For example, the cabbage looper *Trichoplusiani* usually feeds at noon. Plants anticipate this behavior, and JA levels increase early in the day and peak at noon under the control of the circadian system to defend against invasion by these insects [107].

Upon biotic and abiotic stress, ROS are rapidly and massively induced in plants in rhythmic patterns, similar to the patterns of regulation of SA and JA synthesis by the circadian clock, with 39% ROS-related genes displaying these rhythmic expression patterns [108,109]. A few studies have shown that the circadian clock system regulates the expression of these genes through CCA1, LHY, and LUX, thereby influencing the ROS-dependent immune response [110]. In addition, other clock components, such as ELF3, ELF4, TIC, PRR5, PRR7, PPR9, and GI, can also exert gating effects on ROS-related stress responses [95]. Collectively, these various effects of the plant circadian clock play important roles in regulating the innate immune system, which is essential for multi-layered defense responses and broad-spectrum resistance to various pathogens and pests.

The circadian clock also affects root-microbe interactions, which, in turn, have crucial effects on nutrient absorption, ultimately influencing crop yield and biomass. In *Medicago truncatula*, the *lhy* mutant showed reduced root nodule formation, suggesting that the circadian clock may positively regulate the interactions between roots and the associated microbes [111]. Moreover, different bacterial communities exhibit fluctuations in abundance between light and dark cycles, a phenomenon that is partially regulated by CCA1 [112]. In *Arabidopsis*, the rhizosphere communities of *toc1-21* and *ztl-30* mutants are significantly altered compared to those of wild-type plants, suggesting that the circadian clock system impacts the structure and function of these communities [113]. Current evidence favors the idea that the circadian clock functions as a positive effector during root-microbe interactions. However, there are only a few applications of the current knowledge of the circadian clock system for manipulating root-microbe interactions in practical agricultural production.

## 6. Circadian Clock and Metabolic Signals

Rhythmic metabolism is a significant part of the circadian clock system in plants [114,115]. Plant circadian systems anticipate dawn and sunset to optimize carbon fixation [116]. The current evidence supports that the plant circadian clock impacts both primary and secondary metabolism functions in both growth and response to stress, which indicates that the regulatory network of the circadian clock and metabolic signals are potentially important for crop yields [115].

The rhythmic and endogenous sugar signals from photosynthesis can entrain circadian rhythms in Arabidopsis by regulating *PRR7* expression [117]. Metabolic signaling pathways have shown the mechanisms by which sugars can affect clock components. The Snf1-RELATED KINASE1 (SnRK1) and TARGET OF RAPAMYCIN (TOR) kinases are the typical representations of this [118,119].Nutrient ions, such as Ca^2+^, Mg^2+^, and Fe^2+^, are also found to act in regulatory relationships between metabolic signals and the circadian clock [120,121,122]. OsPRR59 and OsPRR95 negatively regulate the rhythmic expression of *OsMGT3*, which encodes a chloroplast-localized Mg^2+^ transporter, thus modulating Mg^2+^ fluctuations. Rhythmic Mg^2+^ fluctuations control carbon fixation and sugar accumulation in rice chloroplasts. Besides, sugar could also trigger superoxide production to positively regulate the expression of *OsPRR95* and *OsPRR59*, which function as a feedback signal to link the circadian clock and metabolic signals [44].

ROS are byproducts in metabolic signals that act as vital retrograde signaling messengers [123,124]. H_2_O_2_ and O_2_^−^ have been found to exert influence on the expression of circadian clock genes [125,126]. Sucrose could promote the expression of *TOC1* in the evening under the involvement of O_2_^−^ [125]. Besides, CCA1 also affects the transcriptional regulation of ROS-responsive genes and ROS homeostasis, thus regulating the tolerance to oxidative stress [126]. The accumulation of ROS leads to the biosynthesis of a series of metabolites, which plants subsequently exploit as retrograde signaling factors to report the metabolic status of the mitochondria and chloroplast [124]. Metabolism is a central part of the circadian system, and most metabolic processes require the participation of the circadian system. The post-translational modifications of metabolism-associated clock proteins, such as *O*-fucosylation, and feedbackloops, such as the retrograde signaling in various organelles, have shown the complexity involved in trying to understand the multiple layers and mutual effects between the circadian clock and metabolism [127,128,129]. The close and complex interrelations between metabolic signals and the circadian system indicate the potential capacity to improve crop yields.

## 7. Conclusions and Perspective

The plant circadian clock plays numerous important roles regarding adaptation to the environment and the determination of crop yields. However, there are still large gaps in our knowledge of the relationship between circadian clock systems and the integration of multiple environmental signals. How do circadian clock systems integrate multiple dynamic environmental signals, such as light and temperature, to control flowering processes? How do circadian clock systems balance the levels of different responses to various stresses simultaneously, for example, in saline or alkaline stress and immunity responses? In the future, the selection and potential modification of circadian loci that change the circadian phase to improve crop adaptation to various temperatures, latitudes, and light and soil conditions may provide a new paradigm for current agricultural practices. A recent study investigated the complexity differences between rhythmically expressed transcripts in Arabidopsis and polyploid crop wheat, which might help us to understand the selection of circadian clock locus during domestication [130].

A deeper understanding of the molecular network underlying the role of the plant circadian clock in biotic stress will be required in order to utilize the circadian knowledge to increase future crop yields. But there are still many puzzles to be resolved. For instance, how do the reverse signals from plant immunity responses to the circadian system reset or entrain the circadian clock to allow for adaptation to stress conditions? How do tissue-specific circadian clocks respond in defense processes? Besides, we might utilize the regulatory relationships of organelle-specific circadian clocks to change the source-to-sink translocation of carbon and nitrogen. Are there differences in the defense mechanism between crop plants and the model plant Arabidopsis? How do different epigenetic modifications co-ordinate and ensure biomass? Finally, how can the regulation of plant resistance by the circadian clock be applied in practical agricultural production to improve crop yield? These questions await further exploration and analysis.

Recently, the concept of chronoculture, also known as agro-chronobiology, was proposed [131,132] with the goal of incorporating our increasing understanding of circadian biology into agricultural production. Accumulated research data from circadian clock studies from a variety of different species have guided the practical application of chronoculture. Chronoculture aims to co-ordinate the circadian rhythms of crop plants and the interacting organisms with farming practices for increasing crop yields. Fully understanding and utilizing circadian clock systems will certainly contribute to the formation of more refined agricultural production systems. In the future, achieving a more insightful understanding of circadian clock systems and the methods for its remolding and utilization to enhance crop yield are exciting fields waiting to be explored.

**Table 1 plants-12-01897-t001:** Function of circadian clock genes in agricultural traits.

Species	Gene Name	Arabidopsis Homolog(s)	Role/Trait	References
*O. sativa*	*OsCCA1* *OsPRR1* *OsPRR37* *OsPRR73* *OsPRR59* *OsPRR95* *OsGI* *OsELF3-1* *OsELF3-2* *OsELF4a* *OsELF4b* *OsELF4c* *OsLUX*	*CCA1* *TOC1* *PRR3/PRR7* *PRR3/PRR7* *PRR5/PRR9* *PRR5/PRR9* *GI* *ELF3* *ELF3* *ELF4* *ELF4* *ELF4* *LUX*	Flowering time and salt stress response regulation Flowering time regulationSalt stress response Photosynthetic carbon fixation Photosynthetic carbon fixation Flowering time and salt stress response regulation Flowering time and salt stress responses regulation Immunity against *M. oryzae* Flowering time and salt stress response regulation Flowering time regulation, salt stress response, and plant immune response Vegetative growth and flowering	[37,81] [41] [41,75] [41,75] [41,44] [41,44] [46,75,133] [44,46] [103] [46,103] [46,134] [46,134] [46,134]
*G.max*	*GmGI* *GmLHY1a* *GmLHY1b* *GmLHY2a* *GmLHY2b* *GmELF3* *GmPRR3a* *GmPRR3b*	*GI* *LHY* *LHY* *LHY* *LHY* *ELF3* *PRR3* *PRR3*	Flowering time regulation and yield determination Drought tolerance and flowering time regulation Drought tolerance and flowering time regulation Flowering time regulation Flowering time regulation Flowering time regulation Flowering time regulation Flowering time regulation	[135] [34,90] [34,90] [34] [34] [32] [34] [34]
*P. sativum*	*LATE1* *HR* *DNE* *SN*	*GI* *ELF3* *ELF4* *LUX*	Flowering time regulation Flowering time regulation Flowering time regulation Flowering time regulation	[136] [137] [137] [137]
*Z. mays*	*ZMGI1* *ZMGI2* *ZmCCA1*	*GI* *GI* *CCA1*	Flowering time regulation Drought response	[138] [138] [139]
*Brassica* *oleracea*	*BoGI*	*GI*	Flowering time regulation, leaf senescence, and post-harvest yellowing retardation	[140]
*Triticum aestivum*	*TaPRR1*	*TOC1*	Heading date, plant height, and thousand-grain weight	[141]

## Figures and Tables

**Figure 1 plants-12-01897-f001:**
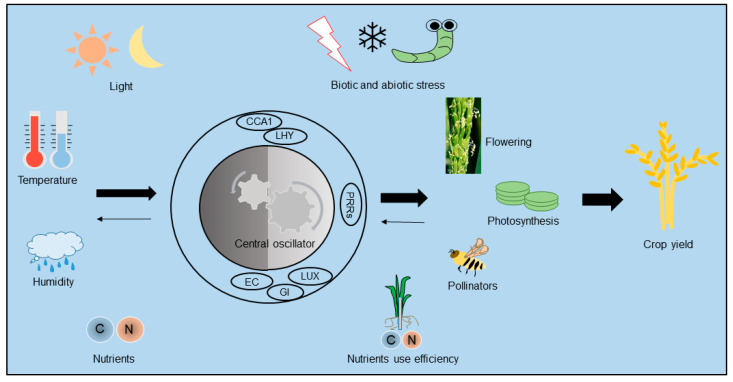
The circadian clock system in a crop, from the input of environmental factors to the determination of crop yield. External environmental factors, including light, temperature, humidity, and nutrients, provide complex input signals for the circadian clock. The core central oscillator of the circadian system receives and converts these signals to enable the plant to adapt to environmental changes, thus affecting flowering, photosynthesis, pollination, nutrient uptake, and responses to biotic and abiotic stresses, all of which ultimately influence crop yields.

**Figure 2 plants-12-01897-f002:**
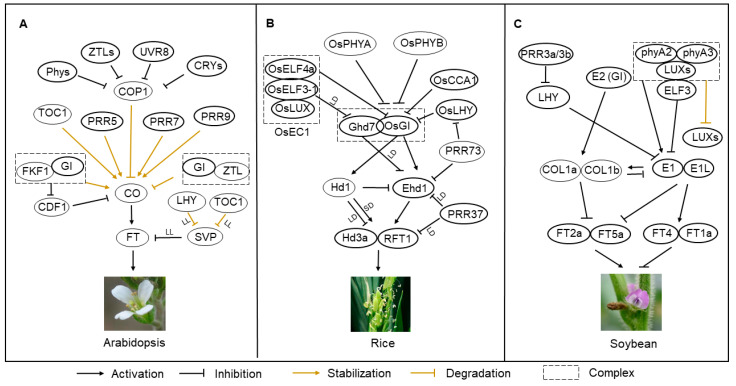
The circadian clock system involved in regulating photoperiodic flowering in Arabidopsis, soybean, and rice. (**A**) In Arabidopsis, the circadian clock components affect the flowering process mainly by regulating CDFs and the GI-CO-FT module transcriptionally and post-transcriptionally. (**B**) The GI-CO-FT signaling pathway is conserved in rice, which is known as the OsGI-Hd1-Hd3a/ RFT1 module. Rice also has some unique components and pathways that modulate the heading date, such as Hd1, Ehd1, and Ghd7, which are also regulated by the circadian clock. (**C**) Similarly, there is a conserved flowering regulation pathway named the GI (E2)-CO-FT module in soybean. Besides, the GI (E2)-CO-E1-FT module also exists in soybean. COP1, CONSTITUTIVELY PHOTOMORPHOGENIC1; FT1a, FLOWERING LOCUS T 1a; COL1a, CO-like gene1a; COL1b, CO-like gene1b; E1L, E1-like gene.

**Figure 3 plants-12-01897-f003:**
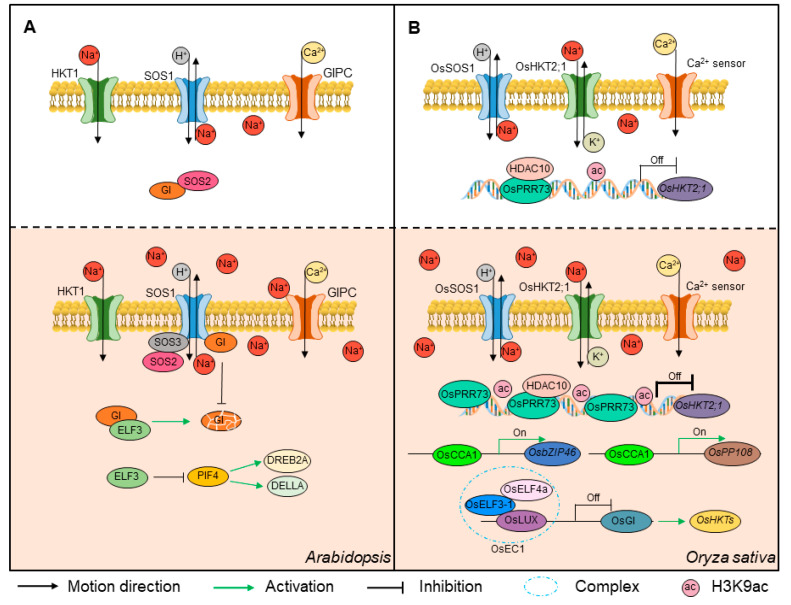
Circadian clock involved in salt stress in Arabidopsis and rice. (**A**) In Arabidopsis, ELF3 interacts with GI and then subsequently degrades GI under salt stress. GI confers salt tolerance based on its degradation and releases SOS2. The SOS2 released recruits SOS3 and forms an active complex that activates SOS1, thus promoting salt tolerance. SOS1 interacts with GI to stabilize GI, conferring salt tolerance in a salt-dependent manner. (**B**) In rice, OsPRR73 directly interacts with HDAC10, thereby repressing the transcription of *HKT2;1* to maintain ion homeostasis. OsCCA1 can bind the promoters of *OsPP2C* and *OsbZIP46* to activate their transcription, thus conferring salt stress tolerance. The OsEC1 complex (including OsELF3-1, OsELF4a, and OsLUX) binds to the promoter of *OsGI* to repress its expression and conveys salt stress through downstream OsHKT2;1/2;3/2;4. GIPC: glycosyl inositol phosphorylceramide.

## Data Availability

Data is contained in the article.

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
