# Peer review of "The Regulatory Networks of the Circadian Clock Involved in Plant Adaptation and Crop Yield"

_plants, 2023, doi:10.3390/plants12091897_

Round 1
Reviewer 1 Report
Please find the attachment with the corrected version of the manuscript.
The topic of the review manuscript "The regulatory networks of circadian clock in plant adaptation and crop yield is interesting, but manuscript is difficult to follow. Moreover, language edition is also required.

Author Response
Response: Sorry for this confusion. We have some minor modifications in manuscript language and logic in the revised version. We also made the changes as you suggested. Thanks!
Reviewer 2 Report
The title of the manuscript promised me an attractive review of circadian clock regulations in plant adaptation and crop yield. That is a daunting task, and its delivery is probably beyond the scope of this manuscript. Diel rhythms and circadian oscillations have been the subject of many recent reviews (e.g., clock adaptation to the environment, 10.1093/plphys/kiac337; clock and stress response, 10.1007/s44154-022-00040-7; metabolites in clock regulation, 10.1016/j.pbi.2022.102333). It will be very difficult to attract attention to this manuscript unless some novel approach or integrative analysis of previous results is provided.
Major issues:
1/ There are typos, and mistakes in grammar and style. The manuscript requires extensive language and style editing. This is a review and it should be well-written.
2/ The authors present a relatively simplistic overview of the circadian clock that is missing many known factors (e.g., metabolic feedback, retrograde signaling, posttranscriptional and posttranslational modifications, and protein turnover).
3/ The previous issue could be justified if the review was dedicated to crops (as indicated in the title) and delivered the promised review of the circadian clock on crop yields. The omission of recently published manuscripts (e.g., 10.1016/j.jplph.2022.153906) is acceptable. However, the absence of references to S. tuberosum published in July 2022 (10.1002/pld3.425) or Brassica oleracea published in 2015 (10.1007/s11105-015-0852-3) indicates that the review was not very thorough.
4/ Figures. The visual site of the two included figures is decent, but the content is confusing and misleading. The depicted abbreviations are not explained, the meaning of dashed boxes in Fig. 1 is not indicated, and the comparison of Arabidopsis with Oryza erroneously indicates that there is no transcriptional control in Arabidopsis (Fig. 2). The depicted processes are not explored at the same level of detail, and the resulting impact is misleading information that there is no overlap in the regulation of flowering.
5/ The promised impact of the circadian clock on crop yield is limited to conclusions and the discussion of chronoculture.
Minor issues:
L25 'circadian clock was evolved in higher plants' circadian clock is not exclusive to higher plants (10.1016/S1937-6448(10)80006-1; 10.1111/pce.14364)
L28 'circadian clock regulates 30% of Arabidopsis genes' - the included reference is seriously outdated dated to 2008
L66 '13 photoreceptors in Arabidopsis' - that statement is misleading, there are only five photoreceptors encoded by multiple genes
L73 Phototropins do participate in the maintenance of circadian oscillations (10.1111/tpj.12947 )
L108 'recent study identified J as an ortholog...' confusing sentence, the J locus abbreviation should be explained
L163-174 Does not provide substantial information
L315-318 - very difficult to follow
Author Response
The title of the manuscript promised me an attractive review of circadian clock regulations in plant adaptation and crop yield. That is a daunting task, and its delivery is probably beyond the scope of this manuscript. Diel rhythms and circadian oscillations have been the subject of many recent reviews (e.g., clock adaptation to the environment, 10.1093/plphys/kiac337; clock and stress response, 10.1007/s44154-022-00040-7; metabolites in clock regulation, 10.1016/j.pbi.2022.102333). It will be very difficult to attract attention to this manuscript unless some novel approach or integrative analysis of previous results is provided.
Major issues:
1/ There are typos, and mistakes in grammar and style. The manuscript requires extensive language and style editing. This is a review and it should be well-written.
Response: Sorry for this confusion. We have revised the manuscript.
2/ The authors present a relatively simplistic overview of the circadian clock that is missing many known factors (e.g., metabolic feedback, retrograde signaling, posttranscriptional and posttranslational modifications, and protein turnover).
Response: Sorry for this confusion. The mainly outer factors were elucidated, and the inner factors, such as metabolic feedback, were basically induced by the change of light and temperature.
3/ The previous issue could be justified if the review was dedicated to crops (as indicated in the title) and delivered the promised review of the circadian clock on crop yields. The omission of recently published manuscripts (e.g., 10.1016/j.jplph.2022.153906) is acceptable. However, the absence of references to S. tuberosum published in July 2022 (10.1002/pld3.425) or Brassica oleracea published in 2015 (10.1007/s11105-015-0852-3) indicates that the review was not very thorough.
Response: Sorry for this confusion. We have revised related parts and added the references.
4/ Figures. The visual site of the two included figures is decent, but the content is confusing and misleading. The depicted abbreviations are not explained, the meaning of dashed boxes in Fig. 1 is not indicated, and the comparison of Arabidopsis with Oryza erroneously indicates that there is no transcriptional control in Arabidopsis (Fig. 2). The depicted processes are not explored at the same level of detail, and the resulting impact is misleading information that there is no overlap in the regulation of flowering.
Response: Sorry for this confusion. We have revised Fig. 1 and Fig. 2.
5/ The promised impact of the circadian clock on crop yield is limited to conclusions and the discussion of chronoculture.
Response: Sorry for this confusion. The promised impact of the circadian clock on crop yield is not directly stated by many papers. However, the agronomic traits that were related to crop yield were exposited in detail.
Minor issues:
L25 'circadian clock was evolved in higher plants' circadian clock is not exclusive to higher plants (10.1016/S1937-6448(10)80006-1; 10.1111/pce.14364)
Response: Sorry for this confusion, we have added this evidence.
L28 'circadian clock regulates 30% of Arabidopsis genes' - the included reference is seriously outdated dated to 2008
Response: Sorry for this confusion, we have revised this part.
L66 '13 photoreceptors in Arabidopsis' - that statement is misleading, there are only five photoreceptors encoded by multiple genes
Response: Sorry for this confusion, we have revised this sentence.
L73 Phototropins do participate in the maintenance of circadian oscillations (10.1111/tpj.12947 )
Response: Sorry for the mistake. We have revised this part.
L108 'recent study identified J as an ortholog...' confusing sentence, the J locus abbreviation should be explained
Response: Followed your suggestion, we have added the explanation of locus J.
L163-174 Does not provide substantial information
Response: Sorry for this confusion. We have revised this part.
L315-318 - very difficult to follow
Response: Sorry for this confusion. We have revised this part.
Round 2
Reviewer 1 Report
Please find the attachment with the corrected version of the manuscript.
The manuscript has been improved after authors correction, however still minor issues has to be addressed, mainly English language-related. Moreover, the abstract should be re-written due too many repeats of one expression.

Author Response
We appreciate your efforts to handle our manuscript. We have further improved our language as the tracked version displayed. We hope you will feel the updated version is acceptable now.
Reviewer 2 Report
The revised manuscript addressed some of my minor issues, but most of my major issues have not been addressed.
Major issues
1/ The language and style of the manuscript still require professional editing.
2/ The authors present a relatively simplistic overview of the circadian clock that is missing many known factors (e.g., metabolic feedback, retrograde signaling, posttranscriptional and posttranslational modifications, and protein turnover).
The authors of a review article should be well-acquainted with the literature, but here are a few examples that could help you in addressing this issue:
10.1111/nph.15525
10.3389/fpls.2019.01614
10.3390/genes12030325
10.3389/fpls.2021.804468
3/ The review has not been updated with relevant references to recent publications on crop production and circadian oscillations, and there are many more published in recent years, e.g., 10.3389/fpls.2020.00285; 10.1371/journal.pbio.3001802
4/ Figures were partially revised. However, the revision did not address the issue. It is impossible to identify evolutionarily conserved mechanisms and pinpoint species-specific regulations. Color coding does not have any visible meaning, and orthologs are depicted in different colors. Abbreviations (including protein names) are not explained.
In other words, I don't see any value in presenting different models if the pathway similarities/differences are not highlighted.
Minor issues
Most minor issues were addressed, but at least this one requires more attention:
The authors have corrected the outdated information about the proportion of genes regulated by circadian/diurnal rhythms. However, the included reference [1] is incorrect. Besides being outdated (2011), it does not contain the referenced information. There is also a mistake in the reference itself. Miguel Blazquéz was the editor of this manuscript, not its first author.
Author Response
The revised manuscript addressed some of my minor issues, but most of my major issues have not been addressed.
Major issues
1/ The language and style of the manuscript still require professional editing.
Response: Following your suggestions, we have further carefully revised our manuscript. We hope you will feel the revised version is improved. Thanks!
2/ The authors present a relatively simplistic overview of the circadian clock that is missing many known factors (e.g., metabolic feedback, retrograde signaling, posttranscriptional and posttranslational modifications, and protein turnover).
The authors of a review article should be well-acquainted with the literature, but here are a few examples that could help you in addressing this issue:
10.1111/nph.15525
10.3389/fpls.2019.01614
10.3390/genes12030325
10.3389/fpls.2021.804468
Response: We have added above evidences in the manuscript. As this review majorly focus on the circadian outputs, we did not include the comprehensive regulatory factors of circadian clock. Thanks for your understanding.
3/ The review has not been updated with relevant references to recent publications on crop production and circadian oscillations, and there are many more published in recent years, e.g., 10.3389/fpls.2020.00285; 10.1371/journal.pbio.3001802.
Response: Thanks for your suggestions,we have updated the suggested new publications in the manuscript.
4/ Figures were partially revised. However, the revision did not address the issue. It is impossible to identify evolutionarily conserved mechanisms and pinpoint species-specific regulations. Color coding does not have any visible meaning, and orthologs are depicted in different colors. Abbreviations (including protein names) are not explained.
In other words, I don't see any value in presenting different models if the pathway similarities/differences are not highlighted.
Response: In this revised version, we have emphasized the evolutionarily conserved modules of flowering regulation and identified the species-specific regulations in different crops. Besides, we added the annotations of some abbreviations in figures and figure legends which were not mentioned in text.
Minor issues
Most minor issues were addressed, but at least this one requires more attention:
The authors have corrected the outdated information about the proportion of genes regulated by circadian/diurnal rhythms. However, the included reference [1] is incorrect. Besides being outdated (2011), it does not contain the referenced information. There is also a mistake in the reference itself. Miguel Blazquéz was the editor of this manuscript, not its first author.
Response: Thanks for your corrections. We have added the above information into the manuscript and corrected the reference information.
Round 3
Reviewer 2 Report
The revised manuscript has been improved, but my concerns have not been addressed.
Language is better but not optimal, and the figures are still misleading.
Here is an example illustrating my point:
Figure 2
Phytochromes are green, a shade of red&yellow, gray&pale green for panels A, B, and C, respectively. LHY is yellow in C, reddish in B, and missing! in panel A.
Unfortunately, similar sloppiness is present in the whole manuscript.
Author Response
Response: Thanks for your corrections. We have deleted the misleading colors in Figure 2 and added some details in figures.